# Temperature and Concentration Dependence of Human Whole Blood and Protein Drying Droplets

**DOI:** 10.3390/biom11020231

**Published:** 2021-02-05

**Authors:** Anusuya Pal, Amalesh Gope, Germano Iannacchione

**Affiliations:** 1Order-Disorder Phenomena Laboratory, Department of Physics, Worcester Polytechnic Institute, Worcester, MA 01609, USA; gsiannac@wpi.edu; 2Department of English, Tezpur University, Tezpur 784028, Assam, India; amalesh@tezu.ernet.in

**Keywords:** drying droplet, colloid, protein, blood, self-assembly

## Abstract

The drying of bio-colloidal droplets can be used in many medical and forensic applications. The whole human blood is the most complex bio-colloid system, whereas bovine serum albumin (BSA) is the simplest. This paper focuses on the drying characteristics and the final morphology of these two bio-colloids. The experiments were conducted by varying their initial concentrations, and the solutions were dried under various controlled substrate temperatures using optical and scanning electron microscopy. The droplet parameters (the contact angle, the fluid front, and the first-order image statistics) reveal the drying process’s unique features. Interestingly, both BSA and blood drying droplets’ contact angle measurements show evidence of a concentration-driven transition as the behavior changes from non-monotonic to monotonic decrease. This result indicates that this transition behavior is not limited to multi-component bio-colloid (blood) only, but may be a phenomenon of a bio-colloidal solution containing a large number of interacting components. The high dilution of blood behaves like the BSA solution. The ring-like deposition, the crack morphology, and the microstructures suggest that the components have enough time to segregate and deposit onto the substrate under ambient conditions. However, there is insufficient time for evaporative-driven segregation to occur at elevated temperatures, as expected.

## 1. Introduction

The whole human blood is a complex bio-colloidal fluid consisting of cellular components [red blood cells (RBCs or erythrocytes), white blood cells (WBCs or leukocytes), and platelets (thrombocytes)] and plasma [1]. On the other hand, the plasma is made of water, and minute amounts of ions, salts, and albumin proteins [2]. The human blood solution, thus, comprises a successive hierarchy levels, representing just one facet of its complexity and various functions. Note that a pure protein solution, while a simple system, can be thought of as the dilute limit of whole blood, which is the most complex bio-fluid.

During the past decade, significant progress has been observed in the drying droplets of bio-colloids, such as [bovine serum albumin (BSA), and lysozyme] protein [3,4,5,6], plasma [7], DNA [8], serum [9], blood [10], etc., due to biomedical and forensic applications [9,11,12,13,14,15]. The coupling between the droplet and the substrate, the surrounding environment, and the constituent particles show a wide variability range [16]. This wide range leads to multiple questions that many researchers attempted to answer in recent years [17,18,19,20]. For example, is the droplet pinned to the substrate? If so, what is the effect on the droplet once it dries up? How does the morphological pattern relate to the initial state of constituent particles? Why does the droplet form cracks? How does the initial contact angle affect the morphology of the droplet? Despite the intense research at each hierarchy level, for example, blood, plasma, and protein, a few studies are conducted to understand such levels’ connection in terms of drying droplets. Manouk et al. [21] have investigated how the blood, plasma, and the plasma’s molecular components (especially BSA) influence the drying evolution and morphological patterns. They concluded that each hierarchy level has a role in deciding the physical mechanism involved during the drying process. Though this work provides preliminary information, many crucial questions were unanswered. For example, does the hierarchy influence the wettability of the droplets? How do the microstructures at each level get affected? How does the droplet at each hierarchy behave in different environmental conditions?

Pal et al. [22] conducted a systematic study on the most complex bio-colloid, blood, to understand the self-assembling mechanisms during the drying process. The study reveals a concentration-driven phase transition when the blood is precisely diluted in de-ionized water. The analysis of three different independent measurements (textural analysis, contact angle measurements, and the morphology) confirmed that the complex combinations of the chemical potentials among the cellular components are responsible for such origin. The progress made by Pal et al. [22] on the drying evolution, and the resulting morphological patterns in the diluted blood samples lead to some consequent exciting questions: what happens to the phase transition when a one-component system replaces the multi-component? What kind of interactions plays a role in this type of transition? Does the wettability of the droplet change for a one-component system? How does it change during the drying process? Does the temperature affect differently for one and multi-component drying droplets?

To address these questions, the present study attempts to explore the drying evolution and final morphology of a complex (whole human blood) and a simple (BSA) droplets by varying the initial concentrations at different controlled substrate temperatures. Bright-field optical microscopy is used to capture images during the drying process and of the final dried films. The wetting properties of these droplets are investigated by measuring their contact angles through the drying process. Finally, scanning electron microscopy is used on the final dried morphology to illuminate structural information. The comparison between these two systems reveals the influence of multiple components’ interactions versus features that depend on single-molecule self-interactions.

## 2. Materials and Methods

A volume of 1 μL whole human blood of a healthy donor contains a few cellular components such as 400 to $500\times 10^{4}$ of RBCs, 0.5 to $1\times 10^{4}$ of WBCs, 14 to $40\times 10^{4}$ of platelets, along with a small amount of proteins, salt ions, etc. [1]. The bovine serum albumin (BSA) present in the cows’ blood is chemically identical to the globular protein present in human blood [23]. Each BSA comprises a molecular mass of ~66.5 kDa, and a prolate ellipsoid shape [24,25].

The anti-coagulant (Na-Citrate) mixed whole human blood (Catalog number 7203706) was purchased from Lampire Biological Laboratories (Campbell, CA, USA). The blood is diluted by adding different volumes of de-ionized water (Millipore, 18.2 MΩ·cm at $\sim 25{\;}^{\circ }$C) to prepare a different range of initial blood concentrations ($\phi_{b}$) of 75, 62, 50, 25, and 12.5% (*v*/*v*). It must be noted that the diluent (de-ionized water) in the whole blood could change the native environment of the blood’s components. It is beyond the scope of this paper if any changes in their structures happen prior to the drying process. Therefore, the images are captured soon after the sample preparation to avoid further complications (if any). The various amounts of BSA protein powder (Catalog no. A2153, Sigma Aldrich, St. Louis, MO, USA) was dissolved in 1 mL of de-ionized water to prepare the initial protein concentration ($\phi_{p}$) of 20, 9, 5, 3, and 1 wt%. It can be argued that the usage of buffered saline instead of the de-ionized water will ensure maintaining the native states of the cellular components in the blood and the BSA particles’ conformations. However, one of the primary concerns of this article is to explore the drying mechanism and its effects on their behavior without adding any external salts.

A volume of ~1 μL of each sample is pipetted on a coverslip (Catalog number 48366-045, VWR, USA) to form a ~2 mm diameter droplet under ambient conditions (room temperature of $\sim 25{\;}^{\circ }$C and relative humidity of ~50%). The coverslip is transferred to the hot stage attached to the bright-field optical microscope (Leitz Wetzlar, Germany) within ~45 s. The temperature controller was set at different temperatures (T) of 25, 35, and $45{\;}^{\circ }$C to run the blood samples’ experiments. Besides these temperatures, T = $55{\;}^{\circ }$C is added for the BSA samples. It is worthwhile to mention that the T should be less than the denaturing temperature ($T_{d}$) of the sample as their structures and functions might get transformed above $T_{d}$. The $T_{d}$ of the human blood and the BSA is $\sim 45{\;}^{\circ }$C [26] and $\sim 65{\;}^{\circ }$C [27].

The time-lapse images were recorded under a 5× magnification every two seconds. An 8-bit digital camera (Amscope MU300) and a fixed resolution of 2048×1536 pixels were used for acquiring those images. The images of the dried samples were captured within 24 h. All of the experiments were repeated three times at each concentration and temperature to ensure the outcomes’ consistency. The samples show high reproducibility.

An image processing technique is adopted (using ImageJ [28]) to conduct the quantitative analysis of the images captured during the drying process. The oval tool in ImageJ was used to select the region of the droplet. The statistical mean and the standard deviation of the images are computed. The radius of the fluid front (*r*) is measured five times at each time (*t*). The time evolution of the averaged radius ($\bar{r}\left(t\right)$) is obtained during the drying process. Finally, the averaged radius of the droplets ($\bar{R}$) is also calculated at each concentration. The parameter, $\bar{r}$(t)/$\bar{R}$, is plotted as a function of time. The averaged ring width ($\bar{w}$) of the dried film is also measured. The crack spacing ($x_{c}$) is computed by drawing a circular line at each crack domain. The detailed procedure can be found in our previous papers [29,30]. The $\bar{w}/\bar{R}$ and ${\bar{x}}_{c}$ are plotted as a function of the initial concentration at each temperature.

The contact angle measurements are done under ambient conditions (temperature of $\sim 25{\;}^{\circ }$C and relative humidity of ~50%) using the contact angle goniometer (Model 90, Ramé-hart Instrument Company, Succasunna, NJ, USA). The scanning electron microscopy (JEOL-7000F, JEOL Inc., Peabody, MA, USA) is used for the microstructural analysis of the dried films. The sputter-coating with a 4 nm layer of gold nanoparticles at the accelerating voltage of 3 kV and the probe current of 5 mA.

## 3. Results

### 3.1. Qualitative Analysis

#### 3.1.1. Whole Human Blood: The Most Complex Bio-Colloid

Figure 1 shows the morphology of the whole human blood droplet dried during 24 h at different substrate temperatures (T) ranging from 25 to $45{\;}^{\circ }$C. The various initial concentrations of blood ($\phi_{b}$) from 100 (undiluted whole blood) to 12.5% (*v*/*v*) (diluted whole blood) are also studied at this temperature range. The whole blood texture changes from dark (at $\phi_{b}$ of 100%) to light gray (at $\phi_{b}$ of 12.5%) as we dilute its initial concentration. A ring-like feature is found at all $\phi_{b}$ at T = $25{\;}^{\circ }$C; however, it can be prominently observed at $\phi_{b}$ of 50 to 12.5% (*v*/*v*). It is to be noted that the ring (or the rim) terminology is the same as the blood’s corona in this study. The ring’s width (*w*) decreases, and the central region’s width increases as $\phi_{b}$ is diluted. Both of these regions are found in all the samples except 100% (*v*/*v*) at T = 35 and $45{\;}^{\circ }$C as well as 75% (*v*/*v*) at T = $45{\;}^{\circ }$C. A uniform dark-gray texture is found in both of these regions for the samples ranging from 100 to 62% (*v*/*v*). The difference in their textures gets visible from 50 to 12.5% (*v*/*v*). The central region becomes lighter than the ring at this range. It indicates that the ring and the central regions are at different heights from 50 to 12.5% (*v*/*v*) [22]. Interestingly, the elevated temperatures (T of 35 and $45{\;}^{\circ }$C) show a sharp-edged ring compared to T = $25{\;}^{\circ }$C, independent of their initial concentration.

The $\phi_{b}$ of 100% (*v*/*v*) at T = $25{\;}^{\circ }$C has four radial cracks. These cracks divide the ring (or the corona) into large-sized domains (a similar behavior is observed in [1,19]). Some of the micro-flaws are noticed in these domains. The crack lines are thick. The number of these cracks is increased as we dilute the sample. It is to be noted that the length of these cracks is directly proportional to the ring’s width. The thick crack lines become thin as the sample is diluted. Looking around the edges at $\phi_{b}$ of 100% (*v*/*v*) suggests that these cracked domains are not firmly attached to the substrate. Some of these domain sheets are moved towards the center, and some are away. These misplaced sheets are profoundly exhibited at T = $45{\;}^{\circ }$C. These dried films’ high magnified images display that these films firmly adhere to the substrate (coverslip) at $\phi_{b}$ of 50 to 12.5% (*v*/*v*). The small-sized random cracks in the central region are only seen at $\phi_{b}$ of 100 to 62% (*v*/*v*) under our study’s present resolution. Surprisingly, the blood samples’ crack morphology does not significantly change at the elevated temperatures (T of 35 and $45{\;}^{\circ }$C).

Figure 2(I,II)a–f examines the drying evolution of the blood droplet at $\phi_{b}$ of 100 and 12.5% (*v*/*v*), respectively, at the substrate temperature (T) of 45 and $25{\;}^{\circ }$C. The first image is captured within ~70 s after the droplet’s deposition on the substrate (Figure 2(I,II)a). The images at 12.5% (*v*/*v*) are lighter than 100% (*v*/*v*) under ambient and elevated temperatures. The first stage of the drying process started when the fluid front moves from the periphery towards the central region exhibited in Figure 2(I,II)b,c. The light gray texture starts appearing as the front moves, which is observed predominantly at 100% (*v*/*v*). However, the change of the texture is not notable at 12.5% (*v*/*v*) as the first captured image is already of light gray shade (Figure 2(I,II)a,b). Soon after this movement, the droplet is found to have two distinct regions, the ring (or the corona) and the central regions at T = $25{\;}^{\circ }$C. Interestingly, the corona’s edge is found to be sharp at T = $45{\;}^{\circ }$C.

The second stage of the drying process marks the propagation of the cracks. The morphological difference between the undiluted and the diluted blood droplets begins from this stage (Figure 2(I,II)c,d). The cracks propagate radially from the center to the periphery in the undiluted blood sample [100% (*v*/*v*)]. In contrast, these cracks move from the periphery to the corona (or the ring) at $\phi_{b}$ = 12.5% (*v*/*v*). Subsequently, the cracked domains get detached from the substrate, whereas the central region still adheres to the substrate at $\phi_{b}$ of 100% (*v*/*v*) and T = $25{\;}^{\circ }$C. The radial cracks propagate throughout the droplet at T = $45{\;}^{\circ }$C, leading to the absence of the central region. The increased number of the micro-flaws and an additional sliding of the sheets are also observed at T = $45{\;}^{\circ }$C (Figure 2(I)e,f).

The third stage involves changing the texture (from the light to the dark gray), the appearance of the micro flaws from these radial cracks, and the widening of these radial cracks. However, this stage is found to be absent at 12.5% (*v*/*v*) [Figure 2(II)e,f)].

#### 3.1.2. Bovine Serum Albumin Protein: The Simplest Bio-Colloid

Figure 3 shows the morphology of the bovine serum albumin (BSA) droplet dried for 24 h at different substrate temperatures (T) of 25 to $55{\;}^{\circ }$C and various initial concentrations ($\phi_{p}$) ranging from 20 to 1 wt%. Unlike the blood sample (Figure 1), all BSA’s dried films show a uniform light gray texture. The “coffee-ring” [31] like behavior is seen at every $\phi_{p}$ [25,32]. These films exhibit a shadowy dark texture around the inner edge of the ring. This texture differentiates the film into two separate regions, the rim (or the ring), and the central regions. It also indicates that the ring is at a higher height than the central region. The shadowy texture is prominent at 20 as compared to 1 wt%. This suggests that their heights’ differences are more profound with the increase of BSA’s initial concentration. A peripheral dark band is noticed at $\phi_{p}=20$ wt% and minimized as we dilute the samples to 1 wt%. Similar to the whole blood dried film (Figure 1), the ring’s width (*w*) is found to be dependent (decreased) as $\phi_{p}$ is diluted from 20 to 1 wt% at every temperature.

Furthermore, the sharp-edged ring is found at elevated temperatures (T from 35 to $55{\;}^{\circ }$C). Unlike the blood samples, the cracks in BSA films are mostly observed in the ring. These cracks intervene from the ring to the central regions at $\phi_{p}$ of 20 and 9 wt%. No cracks are found in the central region from 5 to 1 wt%. Most of the cracks are radial, dividing the ring into large-sized domains. The orthoradial cracks are mainly observed to join the consecutive radial cracks (reported in our paper, [25]). Interestingly, the number of radial cracks in the blood increases, whereas its number reduces as the BSA’s initial concentration is diluted. In contrast, the radial cracks’ length reduces in both the blood and the BSA samples with dilution. Interestingly, some curved flaws are noticed around the corners of these domains at $\phi_{p}$ of 20 wt% and T = $55{\;}^{\circ }$C.

Figure 4(I,II)a–f illustrates the various stages of the BSA droplets’ drying process. The concentrated ($\phi_{p}$ of 20 wt% in Figure 4(I)a–f and the most diluted (1 wt% in Figure 4(II)a–f) samples are selected for studying the drying evolution at T of 55, and $25{\;}^{\circ }$C. After the droplets are pipetted (Figure 4(I,II)a), the fluid front recedes from the periphery towards the central region (Figure 4(I,II)b–d); similar to what is also observed in the blood droplets (Figure 2(I,II)b,c) [22,25]. This indicates that the first stage is a characteristic of a drying bio-colloidal droplet, which does not depend on the constituent particles’ type. Along with the fluid front movement, a prominent dark peripheral band is observed at $\phi_{p}=20$ wt%. No such band is found in 1 wt% at T = $25{\;}^{\circ }$C; however, a thin dark circular line is exhibited at T = $55{\;}^{\circ }$C. The distinct regions (the ring and the central) are noticed as the fluid front movement approaches its end. Simultaneously, the cracks propagate from the periphery towards the central region like the diluted blood droplet at $\phi_{b}$ of 12.5% (*v*/*v*). The cracks mostly appear in all the samples except for $\phi_{p}$ of 1 wt% and T = $25{\;}^{\circ }$C. The long radial cracks generate, and, subsequently, the orthoradial cracks join these radial cracks (Figure 4(I,II)d–f). Some of the curved flaws also start forming at $\phi_{p}$ of 20 wt% and T = $55{\;}^{\circ }$C. The $\phi_{p}$ of 1 wt% at T = $55{\;}^{\circ }$C shows that the cracks do not propagate in the central region, whereas these cracks intervene at 20 wt%. The different observations in the blood and the protein samples are detailed in the next section.

### 3.2. Quantitative Analysis

#### 3.2.1. Drying Evolution of the Complex Bio-Colloid

Figure 5a–f and Figure 6a–f show the drying evolution of the statistical image parameters, the mean and the standard deviation [in arbitrary units, (a.u.)] of the blood droplets at the different initial concentrations ($\phi_{b}$) of 100 to 12.5% (*v*/*v*), respectively. The substrate temperature (T) of $45{\;}^{\circ }$C is presented in the left *y*-axis, and T = $25{\;}^{\circ }$C is displayed in the right *y*-axis. The *x*-axis exhibits the drying times in seconds for both temperatures.

The mean is defined as the sum of the pixel values divided by the number of the pixels [30,33]. The mean shows three distinct phases: a rapid rise follows the initial increase, after which a peak appears and, finally, decreases to reach the saturation stage. The initial rise of the mean is when the fluid front starts moving from the periphery to the central region. When the gray texture starts appearing significantly, the pixel values increase, resulting in the mean rapid rise (Figure 5a,f and Figure 2(I,II)a–f). Soon after reaching the maximum values, the mean begins to decrease when the cracks propagate. The standard deviation (SD) ranges from 100 to 50% (*v*/*v*) at both temperatures starts with a rapid rise, followed by a dip. Finally, it increases to reach saturation. The SD measuring the image texture’s complexity can capture the finer textural details [30,33]. The uniformity of the image after the deposition of the droplet is the highest. The dark gray texture of the image for the range from 100 to 62% (*v*/*v*) signifies the low value of the SD. It follows a rise and a dip when the mean increases. The pixel values increase (the mean expands) in the process of changing the texture; however, the image’s uniformity is reduced (or the complexity is escalated). As soon as the front progresses significantly, it results in a dip of the SD values. The growth of these cracks breaks the uniformity, and the SD shoots up.

Interestingly, the SD starts with a high value and reduces at $\phi_{b}$ of 25 and 12.5% (*v*/*v*) (Figure 6e,f). The uniformity of the images deteriorates due to the presence of the lower pixel values in the light gray textured images at this concentration range. It exhibits a broad dip during the fluid front movement. The crack propagation leads to an increase in their values. In addition, finally, it gets saturated at every $\phi_{b}$ as the drying process ends. It is to be noted that the substrate temperature does not induce any significant changes in their behavior (Figure 5 and Figure 6).

Figure 7(I,II) shows the time evolution of the averaged fluid front radius [$\bar{r}\left(t\right)$] that is normalized with the averaged radius of the droplet ($\bar{R}$) at different diluting concentrations ($\phi_{b}$) ranging from 100 to 12.5% (*v*/*v*). The parameter, $\bar{r}\left(t\right)/\bar{R}$, is plotted at T of $45{\;}^{\circ }$C in Figure 7(I) and $25{\;}^{\circ }$C in Figure 7(II). The $\bar{r}\left(t\right)/\bar{R}$ at $45{\;}^{\circ }$C shows an initial slow linear and a later fast nonlinear regime. The averaged fluid front radius [$\bar{r}\left(t\right)$] is plotted as a function of time at both temperatures. Subsequently, the velocity (or the slope) of the fluid front is analyzed. The initial slope values ($m_{1}$) are calculated when a linear fit is done in the linear regime. The $m_{1}$ is found to be weakly dependent on the initial concentration ($\phi_{b}$). It varies from −1.3 μm $s^{-1}$ to −0.4 μm $s^{-1}$. The negative sign indicates that the radius of the fluid front reduces with the drying time. The final slope values ($m_{2}$) are extracted by doing a linear fit in the nonlinear regime. A strong dependency of $m_{2}$ on $\phi_{b}$ is found. It reduces from −8.8 μm $s^{-1}$ to −1.0 μm $s^{-1}$ as $\phi_{b}$ is diluted [inset of Figure 7(I)]. The average value of $m_{2}$ is −5.4 ± 3.1 μm $s^{-1}$. On the other hand, the parameter $\bar{r}\left(t\right)/\bar{R}$ shows a weak nonlinearity in the later time at T = $25{\;}^{\circ }$C. Therefore, the slope values (m) are determined from the whole range. The average value of m is −0.5 ± 0.2 μm $s^{-1}$, which is weakly dependent on $\phi_{b}$. It is to be noted that the radius of the fluid front could only be measured when it covers ~20% of the whole droplet at $25{\;}^{\circ }$C, whereas it covers ~60% of the total droplet area at $45{\;}^{\circ }$C.

The wettability of these blood droplets at T = $25{\;}^{\circ }$C is examined by measuring the contact angle during the drying process [Figure 8(I,II)]. The contact angle is the angle formed by the interfaces of substrate–blood, blood–air, and substrate–air [2]. It shows the drying evolution of the normalized contact angle (θ(t)/Θ) at different initial concentrations ($\phi_{b}$) ranging from 100 to 62% (*v*/*v*) in Figure 8(I) and from 50 to 12.5% (*v*/*v*) in Figure 8(II). The Θ is determined by a linear extrapolation of the contact angle [θ(t)] varying with time, where it appears linear (till ~200 s). The normalization is done by dividing θ(t) with Θ. The presence of a peak-like feature in Figure 8(I) makes the contact angle variation non-monotonic. The range from 320 to 570 s is highlighted in the inset of Figure 8(I) at $\phi_{b}$ of 75 and 62% (*v*/*v*). The value of 0.05 is added to the θ(t)/Θ of 62% (*v*/*v*) [inset of Figure 8(I)] for the clear visualization. The peak is broad and observed during 200–300 s at $\phi_{b}$ = 100% (*v*/*v*). This peak becomes small and gets delayed as $\phi_{b}$ is diluted. It is found during 400–500 s at 75 and 62% (*v*/*v*). This peak becomes absent, and a monotonic decrease of the contact angle is found from 50 to 12.5% (*v*/*v*) [Figure 8(II)]. Furthermore, the θ(t) is plotted as a function of time, and the velocity (or the slope) of the contact angle is measured. The slope for 25 and 12.5% (*v*/*v*) is found to be −0.1351 ± $0.0008^{\circ }\;s^{-1}$ (with *R*${}^{2}$ = 0.994) and −0.1971 ± $0.0012^{\circ }\;s^{-1}$ (with *R*${}^{2}$ = 0.996), respectively. Its increased value from ~0.14 to $\sim 0.20^{\circ }\;s^{-1}$ suggests that the angle gets steeper as $\phi_{b}$ is diluted. The negative sign indicates that the angle reduces with time.

A comparison between Figure 7(II) and Figure 8(I,II) reveals that both the contact angle and the fluid front radius move simultaneously at all $\phi_{b}$ except 12.5% (*v*/*v*) initially. It is to be noted that $\bar{r}\left(t\right)/\bar{R}$ is computed up to ~0.6 [shown in Figure 7(II)], which means that the front still moves with the time until it becomes zero. In contrast, no simultaneous movement of the contact angle and the radius is found at $\phi_{b}$ = 12.5% (*v*/*v*). The contact angle reaches a minimum constant value, and, subsequently, the fluid front changes.

#### 3.2.2. Drying Evolution of the Simplest Bio-Colloid

Figure 9(I,II) shows the drying evolution of the normalized contact angle (θ(t)/Θ) in the BSA droplets at the different initial concentration ($\phi_{p}$) of 20 wt% in Figure 9(I) and from 9 to 1 wt% in Figure 9(II). It shows a non-monotonic decrease with a small peak at 20 wt% [Figure 9(I)]. The peak occurs from ~350 to ~450 s during the contact angle movement. It is found that this duration is the time when the fluid front moves from the periphery towards the central region [Figure 4(I)a–f]. The monotonic decrease (without the presence of any peak) is found at $\phi_{p}$ of 9 to 1 wt% (also reported in [25]). The θ(t) is plotted as a function of time, and the velocity (or the slope) of the contact angle is measured. The slope for 9, 5, and 1 wt% is found to be −0.0731 ± $0.0002^{\circ }\;s^{-1}$ (with *R*${}^{2}=0.998$), −0.0876 ± $0.0004^{\circ }\;s^{-1}$ (with *R*${}^{2}=0.996$), and −0.0674 ± $0.0001^{\circ }\;s^{-1}$ (with *R*${}^{2}=0.999$), respectively. Interestingly, the contact angle varies monotonically for both the blood droplets at $\phi_{b}$ of 50 to 12.5% (*v*/*v*) and the BSA droplets at $\phi_{p}$ of 9 to 1 wt%. However, their slope values differ, suggesting that the variation is steeped in the blood rather than the BSA droplets.

Figure 10(I,II) shows the time evolution of the normalized fluid front radius [$\bar{r}\left(t\right)/\bar{R}$)] at $\phi_{p}$ ranging from 20 to 1 wt% in the BSA droplets at a different substrate temperature (T) of $55{\;}^{\circ }$C in Figure 10(I) and $25{\;}^{\circ }$C in Figure 10(II). The slope values (*m*) are calculated from the whole range by plotting the time evolution of the $\bar{r}\left(t\right)$. The variation of *m* with $\phi_{p}$ is exhibited in the insets of Figure 10(I,II). The *m* decreases from −4.6 μm $s^{-1}$ to −1.4 μm $s^{-1}$ at T = $55{\;}^{\circ }$C. The average value of *m* is −1.3 ± 0.3 μm $s^{-1}$ at T = $25{\;}^{\circ }$C.

The mapping of the contact angle [Figure 9(I,II)] and the fluid front radius [Figure 10(II)] of the BSA droplets reveals that all $\phi_{p}$ show a similar trend, unlike blood droplets. Only the contact angle reduces without any fluid front movement in the initial ~200 s, whereas the simultaneous movement of both the contact angle and the fluid front is found afterward.

#### 3.2.3. Morphological Patterns of the Dried Films

Figure 11(I–III) shows the normalized ring width ($\bar{w}/\bar{R}$) of the blood film as a function of $\phi_{b}$ at T of 45 to $25{\;}^{\circ }$C. The solid line depicts the linear fit over the whole range from 100 to 12.5% (*v*/*v*). The dotted and dash-dotted lines show the linear fit from 100 to 62, and 62 to 12.5% (*v*/*v*). It is to be noted that the linear fits for the range from 100 to 12.5% and from 100 to 62% (*v*/*v*) are not possible at T = $45{\;}^{\circ }$C. Therefore, a dashed line is drawn for the guide to our eyes. The slope values were extracted for the different ranges of $\phi_{b}$. The slope for the range 100 to 12.5, 100 to 62, and 62 to 12.5% (*v*/*v*) is denoted by *m*, $m_{1}$, and $m_{2}$. The *m* is found to be 0.0096 ± 0.0003
$\phi_{b}^{-1}$ (*R*${}^{2}=0.995$) and 0.0054 ± 0.0006
$\phi_{b}^{-1}$ (*R*${}^{2}=0.936$), respectively at T = 35 and $25{\;}^{\circ }$C. On the other hand, the $m_{1}$ is 0.0123 ± 0.0002
$\phi_{b}^{-1}$ (*R*${}^{2}=0.999$), and 0.0089 ± 0.0021
$\phi_{b}^{-1}$ (*R*${}^{2}=0.891$), respectively, at T = 35 and $25{\;}^{\circ }$C. In addition, $m_{2}$ is 0.0094 ± 0.0020
$\phi_{b}^{-1}$ (*R*${}^{2}=0.873$), 0.0067 ± 0.0006
$\phi_{b}^{-1}$ (*R*${}^{2}=0.999$), and 0.0051 ± 0.0008
$\phi_{b}^{-1}$ (*R*${}^{2}=0.928$), respectively, at T = 45, 35, and $25{\;}^{\circ }$C. The higher *R*${}^{2}$ value of *m* compared to that of $m_{1}$ and $m_{2}$ indicates that the linear fit is better suited for the whole range of $\phi_{b}$ instead of splitting the fits at 62% (*v*/*v*).

The variation of the averaged crack spacing (${\bar{x}}_{c}$) of the blood film as a function of $\phi_{b}$ at 45, 35, and $25{\;}^{\circ }$C is presented in Figure 12(I–III), respectively. Similar to Figure 11(I–III), the slope values (*m*, $m_{1}$, and $m_{2}$) are extracted for various concentration ranges. The *m* is found to be 0.0056 ± 0.0006 mm $\phi_{b}^{-1}$ (*R*${}^{2}=0.942$) at T = $45{\;}^{\circ }$C. As we decrease the temperature to $35{\;}^{\circ }$C, the *m* reduces and is observed to be 0.0049 ± 0.0013 mm $\phi_{b}^{-1}$ (*R*${}^{2}=0.721$). Finally, the *m* becomes 0.0034
±0.0012 mm $\phi_{b}^{-1}$ (*R*${}^{2}=0.579$) at room temperature of $25{\;}^{\circ }$C. In contrast, the $m_{1}$ is 0.0081 ± 0.0008 mm $\phi_{b}^{-1}$ (*R*${}^{2}=0.982$), and 0.0081 ± 0.0015 mm $\phi_{b}^{-1}$ (*R*${}^{2}=0.934$) at the elevated temperatures, 45 and $35{\;}^{\circ }$C, respectively. At T = $25{\;}^{\circ }$C, the $m_{1}$ is 0.0011 ± 0.0015 mm $\phi_{b}^{-1}$ (*R*${}^{2}=0.960$). The $m_{2}$ is 0.0039 ± 0.0005
$\phi_{b}^{-1}$ (*R*${}^{2}=0.958$), 0.0029 ± 0.0007
$\phi_{b}^{-1}$ (*R*${}^{2}=0.844$), and 0.0016 ± 0.0005
$\phi_{b}^{-1}$ (*R*${}^{2}=0.736$), respectively, at T = 45, 35, and $25{\;}^{\circ }$C. Unlike the $\bar{w}/\bar{R}$, *R*${}^{2}$ values of $m_{1}$ and $m_{2}$ at all temperatures increases compared to *m* when ${\bar{x}}_{c}$ is plotted as a function of $\phi_{b}$. The slope values change for different ranges of the linear fits done from 100 to 62, and 62 to 12.5% (*v*/*v*). A kink is observed at 62% (*v*/*v*) in all temperatures; however, it reduces with the increase of the temperature [oval line in Figure 12(I–III)].

Figure 13(I,II) shows the variation of the normalized ring width ($\bar{w}/\bar{R}$) and the averaged crack spacing (${\bar{x}}_{c}$), respectively, as a function of $\phi_{p}$ in the BSA film at T ranging from 55 to $25{\;}^{\circ }$C. The green solid line depicts that the width of the ring decreases with the dilution. The slope values of $\bar{w}/\bar{R}$ at T = 55, 45, 35, and $25{\;}^{\circ }$C are 0.033 ± 0.004
$\phi_{p}^{-1}$ (*R*${}^{2}=0.960$), 0.029 ± 0.006
$\phi_{p}^{-1}$ (*R*${}^{2}=0.893$), 0.033 ± 0.007
$\phi_{p}^{-1}$ (*R*${}^{2}=0.866$), and 0.026 ± 0.004
$\phi_{p}^{-1}$ (*R*${}^{2}=0.945$), respectively. These values indicate that the $\bar{w}/\bar{R}$ is independent on the temperature. In contrast, the ${\bar{x}}_{c}$ exhibits a strong dependency on the temperature. The different colored dashed lines guide us to understand the variation of the ${\bar{x}}_{c}$ with the substrate temperature. The straight line of ${\bar{x}}_{c}$ suggests that it is nearly independent of $\phi_{p}$ at T = $55{\;}^{\circ }$C. As T decreases from 55 to $25{\;}^{\circ }$C, the ${\bar{x}}_{c}$ reduces from ~0.45 mm to ~0.21 mm.

## 4. Discussion

Figure 14(I,II)a–c and Figure 15(I,II)a–c show the physical mechanism of both the blood and BSA droplets (please refer to our previous publications [22,25] for further details in this regard). The capillary flow (Figure 14(I,II)a and Figure 15(I,II)a) drives the particles towards the droplet periphery. As time progresses, a BSA-rich layer is observed to form in the one-component system (aqueous BSA solution in (Figure 15(I,II)b). On the other hand, a plasma protein-rich layer develops in the multi-component system (diluted whole blood droplets in Figure 14(I,II)b). With the advancement of time, more water starts evaporating from these droplets, leading to an interaction of the droplet particles. The BSA particles interact in the protein droplets, whereas the cellular components (WBCs, RBCs, and platelets) (significantly) interact in the blood droplets. We assumed that many of these particles and their interactions near the periphery disturb the contact angle measurements. For example, the non-monotonic decrease of the contact angle is observed at $\phi_{b}$ of 100 to 62% (*v*/*v*) in blood droplets and at $\phi_{p}$ of 20 wt% in the BSA droplets [Figure 8(I) and Figure 9(I)]. Please refer to [22] for a detailed discussion on the origin of such non-monotonic property. The drying mechanism of the one component (simplest) and the multi-component (complex) bio-colloidal droplets discussed in this paper firmly establishes the fact that the non-monotonic property (i.e., the presence of the peak-like feature) is indeed a universal property that can be observed across any high-concentrated colloidal samples. A similar observation (i.e., the presence of such non-monotonic property) has recently been reported in the polymeric system [34] that further supports our claims.

A comparison amongst the macroscopic images of the drying process (Figure 2(I,II)a–f and Figure 4(I,II)a–f) confirms that the peak-like feature obtained from the contact angle measurements (non-monotonic property) appears only during the fluid front movement. This indicates that a sufficient amount of water is still present in the droplet to preserve these particles’ native states. Both these droplets, however, experience mechanical stress as soon as the water reduces significantly. This (mechanical) stress affects the native states of RBCs and WBCs and alters the platelets’ biological activity in the blood droplet. The microstructural analysis of the dried films at 100% (*v*/*v*) reveals a uniform texture (also reported in [19,22]) in the ring (marked with a black arrow in Figure 14(I)c). The RBC membranes create this uniform layer, and some deformed RBCs are observed in the cracked peripheral region (indicated with a white arrow in Figure 14(I)c). The central region is crammed with mostly WBCs (the ridges and troughs-like structures) and the platelets (the spread structure with extended filopodia). Some spicules-shaped RBCs are found near the ring’s inner edge at 12.5% (*v*/*v*) (within the orange rectangle in Figure 14(II)c). The sickle-shaped RBCs and the discoid-shaped inactivated platelets are observed in the central region (marked with the blue rectangle in Figure 14(II)c). Please refer to [22] for a detailed microstructural discussion. Interestingly, BSA dried films’ micrographs show a uniform texture (Figure 15(I,II)c). This indicates that these protein structures are at a shallow length scale (unlike blood dried films), and the magnification is not enough to analyze at those length scales.

It is worth mentioning that the morphological texture and the (crack) patterns significantly affect the blood droplets (Figure 1 and Figure 2), but not the BSA droplets (Figure 3 and Figure 4). The sliding of the cracked films is observed at $\phi_{b}$ of 100 to 62% (*v*/*v*); it is, however, absent at $\phi_{b}$ of 50 to 12.5% (*v*/*v*) [Figure 1, reported in [22]]. On the other hand, all the samples of BSA show that the dried films firmly adhere to the substrate (Figure 3, reported in [25]). Contrary to these blood samples, the ${\bar{x}}_{c}$ is found to have a weak concentration dependence in the BSA films (Figure 13(II)). The crack patterns are similar in all the protein samples, which is possibly due to the absence of different-sized components. Furthermore, the ring’s height is higher than the central region at $\phi_{p}=20$ wt%, and it reduces as we dilute $\phi_{p}$ to 1 wt% (Figure 15(I,II)c). The corona region of the blood film at the highest dilution [$\phi_{b}$ of 12.5% (*v*/*v*)] nearly mimics the homogeneous texture of the BSA blood films [Figure 14(II)c and Figure 15(II)c]. The dried films of both simplest and complex bio-colloidal films show a gradual decrease when the ring’s normalized width is plotted with the initial concentration [Figure 11(I–III) and Figure 13(I)]. This trend indicates that the width of the ring measures the deposition of the particles. When the number of particles is high, the ring’s width is more (reported in one of our papers in [25]). As the number reduces, the ring’s width decreases; however, the self-assembling mechanism is different during their deposition. The different self-interacting mechanism (of a large number of components) is observed for other complex multi-component systems [29,32,33], not specifically to the blood.

We really appreciate the idea proposed by one of the reviewers to changing the ratio of blood components and plasma proteins by decreasing RBCs’ concentration using the centrifugation method instead of diluting it with de-ionized water. This particular method is used to study the morphological alteration of RBCs recently [35]. While we understand that a systematic study is required to understand how each cellular component behaves and contributes to the drying process, we must admit that this is beyond the scope of the current paper. The visco-elastic to elastic transition is developed during the drying process when the concentration of the blood components increases. Therefore, starting with the different initial concentrations, i.e., changing the ratio of blood components and plasma proteins, will affect the transition concentration reported in this manuscript. We look forward to examining those aspects in our future work.

In addition to the concentration study, the substrate temperature (T) is varied. The sample is pipetted on the substrate under ambient conditions. The substrate is transferred to the hot stage within ~45 s. This means that a temperature gradient forms between the apex of the droplet and the fluid–substrate interface. This gradient can generate thermally induced Marangoni flow in addition to the capillary flow in the droplets at the elevated temperatures. The images captured during the drying process and the dried films of both the BSA and the whole blood show that the morphological (crack) patterns are not significantly different from those captured under the ambient temperature (Figure 1, Figure 2, Figure 3 and Figure 4). The statistical mean and the standard deviation extracted from these images confirm that the temperature only reduces the drying time (Figure 5 and Figure 6). The fluid front measurements indicate that the fluid’s velocity gets increased with the increase of the temperature [Figure 7(I,II) and Figure 10(I,II)]. The mean crack spacing (${\bar{x}}_{c}$) shows a gradual decrease in the BSA samples [Figure 13(II)]. A kink at 62% (*v*/*v*) gets reduced as the temperature is increased in blood samples [Figure 12(I–III)]. The sharp-edged “coffee-ring” ring is predominantly observed in all the dried films at elevated temperatures [Figure 11(I–III) and Figure 13(I)]. These observations suggest that the particles in both the blood and the BSA droplets have enough time to segregate and deposit their particles onto the substrate under ambient conditions. There is insufficient time for them to segregate fully. Despite having the same stages, the accelerated drying at the elevated temperatures exhibits a weak influence of the thermal Marangoni flow, which is partially counteracted by the particles’ enhanced edge aggregation in such droplets.

It will also be interesting to examine the effects of the relative humidity and/or its coupling with the temperature on the pattern formation of such diluted bio-colloidal droplets since relative humidity directly affects the drying mechanism.

## 5. Conclusions

This paper explored the physical phenomena of different patterns formed during different phases of the drying process in one of the most complex (whole human blood) and a simplest (aqueous solution of the globular protein, BSA) bio-colloidal droplets. The captured images, the statistical image analysis, the fluid front, the contact angle measurements during the drying process, and the dried films’ microstructural sketches are described for both the blood and the BSA samples. These indicate that the samples’ visco-elasticity depends on the self-interacting mechanism of large number of components, not specifically to the blood. The sample prepared by extreme dilution of the whole blood behaves like the BSA droplet. The substrate temperature only increases the rate of the drying process, and does not influence the states of the blood’s components. This current study therefore initiates a theoretical effort to examine how the dilution and the substrate temperature interplay in characterizing these bio-colloidal droplets. This study offers new macroscopic and microscopic insights connecting the hierarchical structures that are evolved as the drying process advances through a new phase transition mechanism.

## Figures and Tables

**Figure 1 biomolecules-11-00231-f001:**
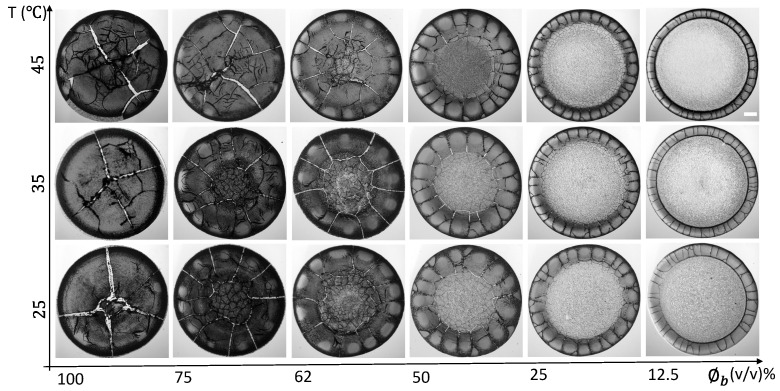
Morphology of the whole human blood film during 24 h at different initial concentrations ($\phi_{b}$) ranging from 100 to 12.5% (*v*/*v*) that are dried at various substrate temperatures (T) of 45, 35, and $25{\;}^{\circ }$C. The scale bar of length 0.2 mm is represented by the white rectangle in the top-right panel.

**Figure 2 biomolecules-11-00231-f002:**
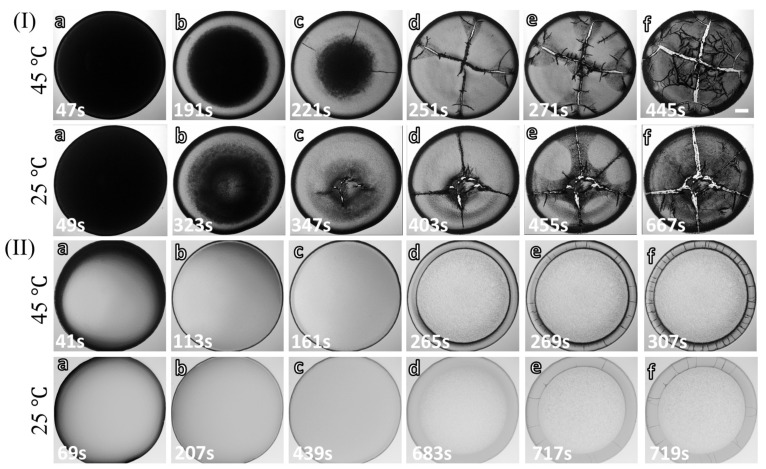
Time evolution of the blood droplets during the drying process: (**a**–**f**) display the drying stages at the substrate temperature (T) of 45 and $25{\;}^{\circ }$C at the initial concentration ($\phi_{b}$) of 100% (*v*/*v*) in (**I**) and 12.5% (*v*/*v*) in (**II**). The scale bar of length 0.2 mm is represented with the white rectangle in the top-right panel.

**Figure 3 biomolecules-11-00231-f003:**
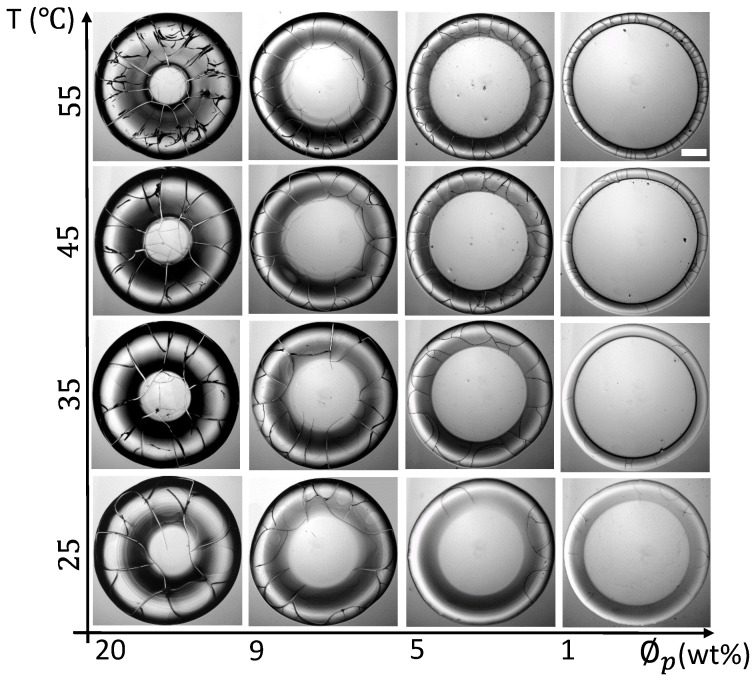
BSA film’s morphology during the 24 h at different initial concentrations ($\phi_{p}$) ranging from 20 to 1 wt% dried at various substrate temperatures (T) of 55, 45, 35, and $25{\;}^{\circ }$C. The scale bar of length 0.3 mm is represented with the white rectangle in the top-right panel.

**Figure 4 biomolecules-11-00231-f004:**
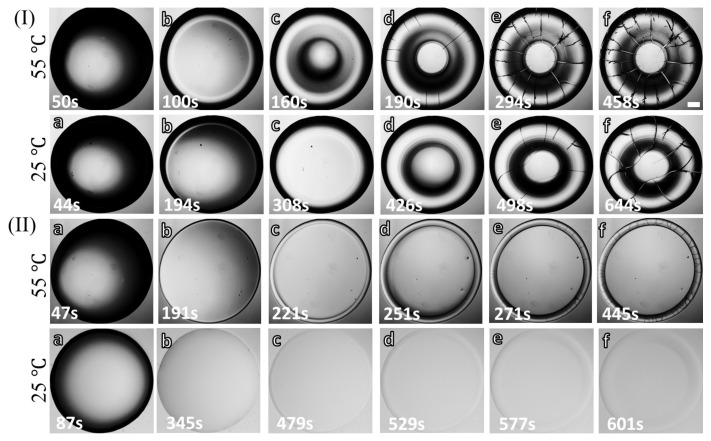
Time evolution of BSA droplets at different substrate temperature (T) during the drying process: (a–f) display different stages at the initial concentration ($\phi_{p}$) of (**I**) 20 wt% for T of 55 and $25{\;}^{\circ }$C, and (**II**) 1 wt% at T of 55 and $25{\;}^{\circ }$C. The scale bar (0.2 mm length) is shown with the white rectangle in the top-right panel.

**Figure 5 biomolecules-11-00231-f005:**
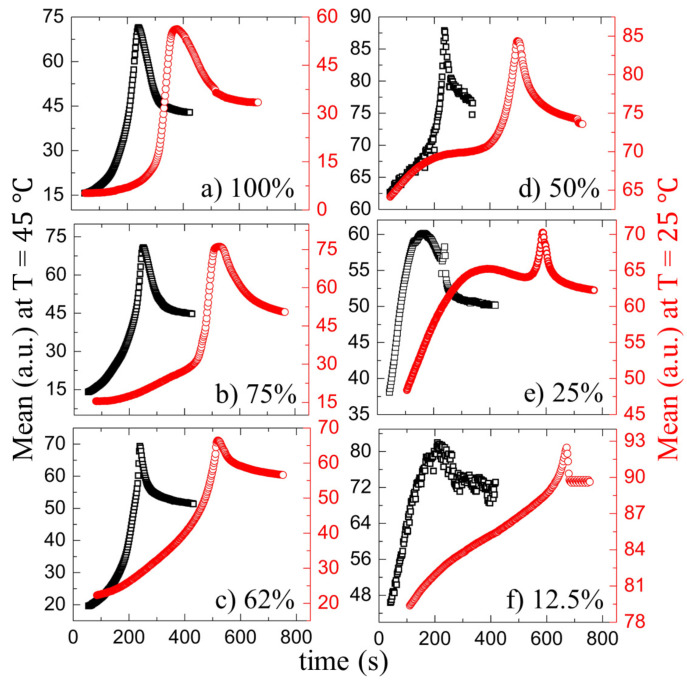
Time evolution of the statistical image parameter, the mean [in arbitrary units, (a.u.)] of the blood droplets at the different initial concentrations ($\phi_{b}$) of 100 to 12.5% (*v*/*v*), respectively, in (**a**–**f**). The black and the red colors represent the mean at 45 and $25{\;}^{\circ }$C respectively.

**Figure 6 biomolecules-11-00231-f006:**
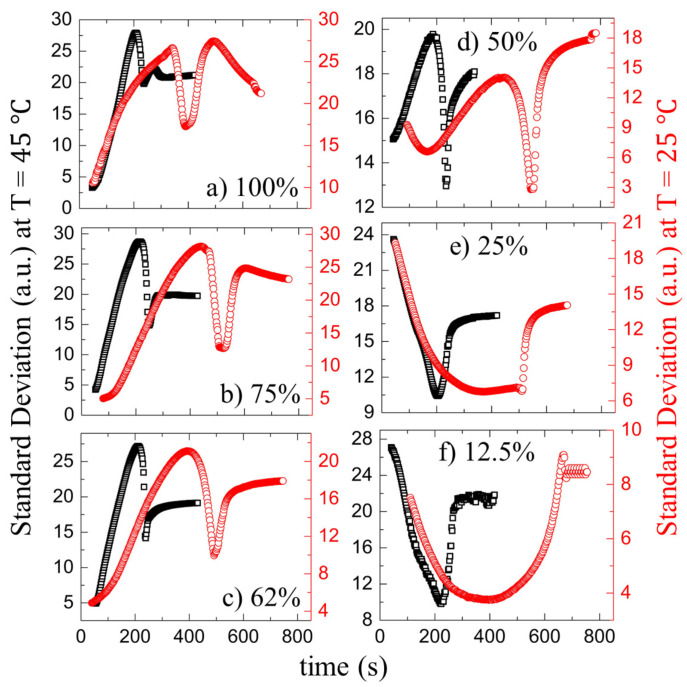
Time evolution of the statistical image parameter, the standard deviation [in arbitrary units, (a.u.)] of the blood droplets at the different initial concentrations ($\phi_{b}$) of 100 to 12.5% (*v*/*v*), respectively, in (**a**–**f**). The black and the red colors represent the standard deviation at 45 and $25{\;}^{\circ }$C respectively.

**Figure 7 biomolecules-11-00231-f007:**
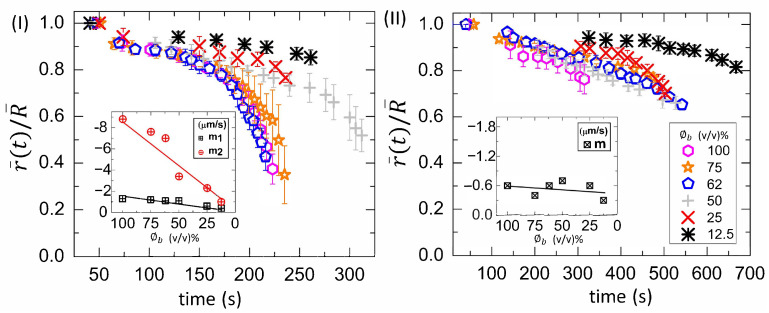
Time evolution of the normalized fluid front radius ($\bar{r}\left(t\right)/\bar{R}$) at different initial concentration ($\phi_{b}$) ranging from 100 to 12.5% (*v*/*v*) in the whole blood droplets at various substrate temperatures (T) of $45{\;}^{\circ }$C in (**I**) and $25{\;}^{\circ }$C in (**II**). The initial ($m_{1}$) and final ($m_{2}$) slope values are calculated from ($\bar{r}\left(t\right)$) measurements in each region at T = $45{\;}^{\circ }$C, whereas the slope values (*m*) are determined from the whole range at T = $25{\;}^{\circ }$C. The variation of the slope with $\phi_{b}$ is displayed in the insets of (**I**) and (**II**).

**Figure 8 biomolecules-11-00231-f008:**
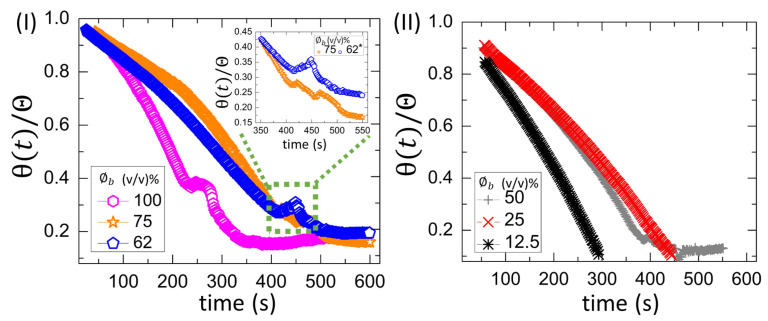
Time evolution of the normalized contact angle (θ(t)/Θ) in the whole blood droplets for T = $25{\;}^{\circ }$C at different initial concentrations ($\phi_{b}$) ranging from 100 to 62% (*v*/*v*) in (**I**) and from 50 to 12.5% (*v*/*v*) in (**II**). The range from 320 to 570 s is zoomed in the inset of (**I**) at $\phi_{b}$ of 75 and 62% (*v*/*v*). The value of 0.05 is added to θ(t)/Θ of 62% (*v*/*v*) (illustrated with an asterisk mark) for the clear visualization.

**Figure 9 biomolecules-11-00231-f009:**
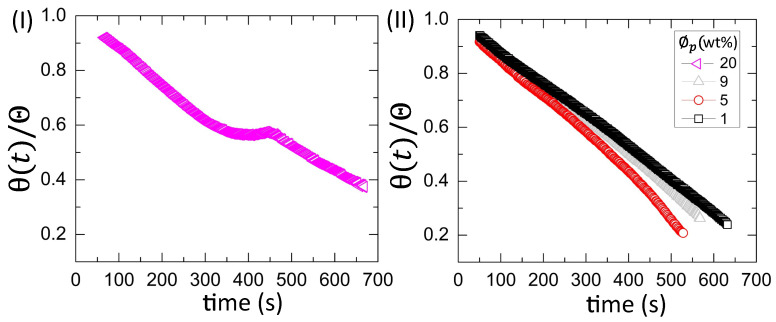
Time evolution of the normalized contact angle (θ(t)/Θ) in the BSA droplets at the different initial concentrations ($\phi_{p}$) of 20 wt% in (**I**) and from 9 to 1 wt% in (**II**) for T = $25{\;}^{\circ }$C.

**Figure 10 biomolecules-11-00231-f010:**
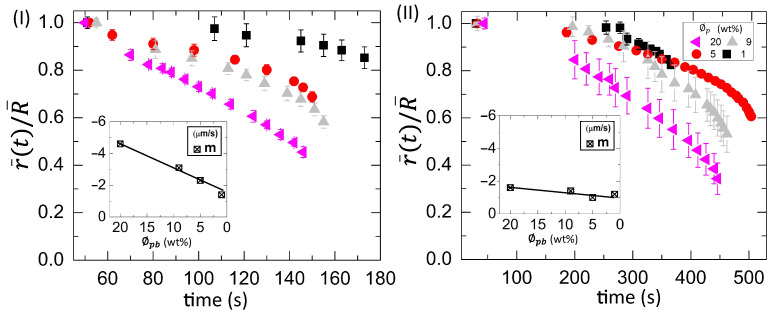
Time evolution of the averaged fluid front radius ($\bar{r}\left(t\right)$) that is normalized with the averaged radius of the droplet ($\bar{R}$) at the different initial concentration ($\phi_{p}$) ranging from 20 to 1 wt% in the BSA droplets at different substrate temperature (T) of $55{\;}^{\circ }$C in (**I**) and $25{\;}^{\circ }$C in (**II**). The slope values (*m*) are calculated from the whole range by plotting the $\bar{r}\left(t\right)$ vs. time. The variation of *m* with $\phi_{p}$ is exhibited in the insets of (**I**) and (**II**).

**Figure 11 biomolecules-11-00231-f011:**
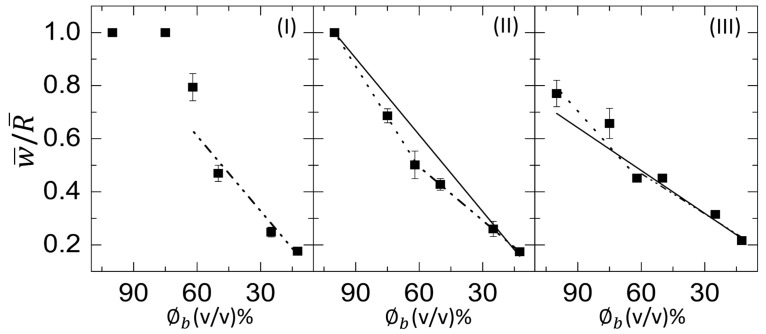
Concentration dependence of the normalized ring width ($\bar{w}/\bar{R}$) of the whole human blood film at different substrate temperatures (T) at (**I**) $45{\;}^{\circ }$C, (**II**) $35{\;}^{\circ }$C, and (**III**) $25{\;}^{\circ }$C. The solid line depicts the linear fit over the whole range from 100 to 12.5% (*v*/*v*). The dotted and dash-dotted lines show the linear fit from 100 to 62%, and 62 to 12.5% (*v*/*v*), respectively. The dashed line at T = $45{\;}^{\circ }$C is a line drawn to guide our eyes.

**Figure 12 biomolecules-11-00231-f012:**
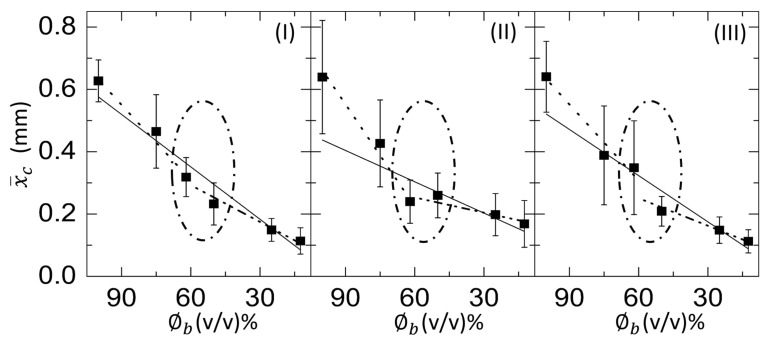
Concentration dependence of the averaged crack spacing (${\bar{x}}_{c}$) of the whole human blood film at different substrate temperatures (T) at (**I**) $45{\;}^{\circ }$C, (**II**) $35{\;}^{\circ }$C, and (**III**) $25{\;}^{\circ }$C. The solid green line depicts the linear fit done over the whole range from 100 to 12.5%. The dotted and dash-dotted lines show the linear fit from 100 to 62, and 62 to 12.5% (*v*/*v*). The oval line highlights a kink at 62% (*v*/*v*).

**Figure 13 biomolecules-11-00231-f013:**
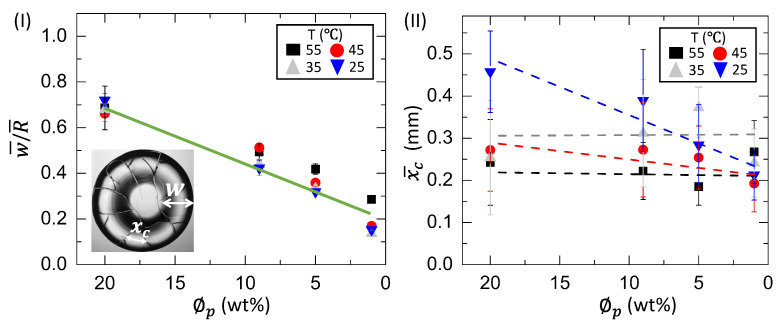
(**I**) Concentration dependence of the normalized width of the ring ($\bar{w}/\bar{R}$) of the BSA film at different substrate temperatures (T) from 55 to $25{\;}^{\circ }$C. The green solid line shows a master linear fit. (**II**) Concentration dependence of the averaged crack spacing (${\bar{x}}_{c}$) of the BSA film at different T from 55 to $25{\;}^{\circ }$C. The different colored dashed lines at different temperatures are drawn to guide our eyes. The $x_{c}$ and *w* are shown on the dried BSA film at $\phi_{p}$ of 20 wt% and T = $25{\;}^{\circ }$C.

**Figure 14 biomolecules-11-00231-f014:**
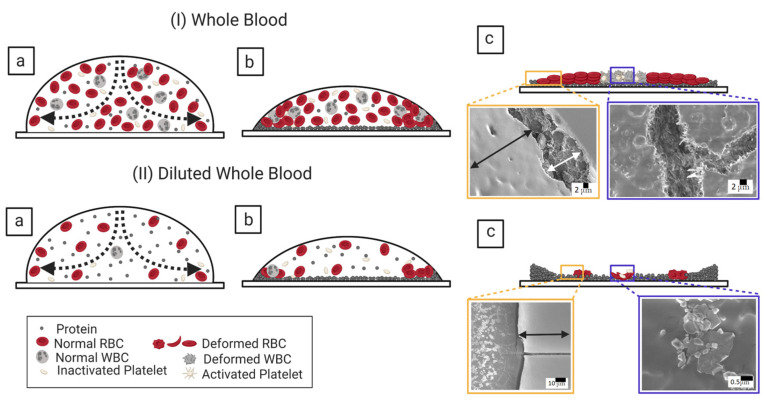
Self-assembling mechanism of the undiluted and the diluted blood droplets during the drying process. The $\phi_{b}$ of 100% (*v*/*v*) is displayed in ((**I**)a–c), and the 12.5% (*v*/*v*) is exhibited in ((**II**)a–c). The dotted arrows in ((**I**,**II**)a) indicates the capillary flow of the fluid. The blue rectangles in ((**I**,**II**)c) display the central region’s microstructures for both 100 and 12.5% (*v*/*v*). The orange rectangle in ((**I**)c) illustrates the periphery and the corona (or the ring) regions at $\phi_{b}$ = 100% (*v*/*v*). The corona and the central regions are presented within the orange rectangle in ((**II**)c). The white arrow reveals the peripheral region, whereas the black arrow demonstrates the ring.

**Figure 15 biomolecules-11-00231-f015:**
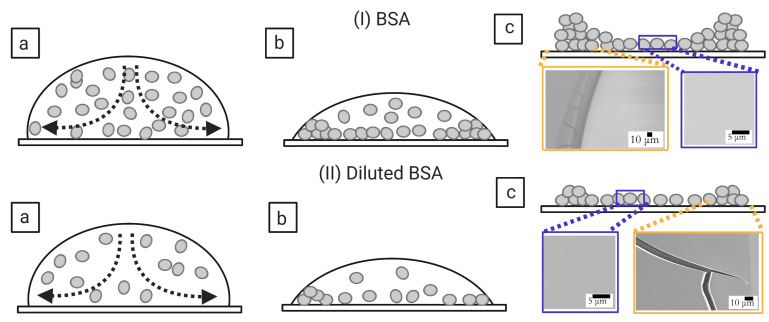
The BSA droplet’s physical mechanism during the drying process at $\phi_{p}$ = 20 wt% in ((**I**)a–c) and at $\phi_{p}=1$ wt% in ((**II**)a–c). The capillary flow is indicated with the dotted arrows in ((**I**,**II**)a). The blue and the orange rectangles in ((**I**,**II**)c) display the microstructures in the central and the rim regions (both 20 and 1 wt%).

## Data Availability

Not applicable.

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
