# Peer review of "Temperature and Concentration Dependence of Human Whole Blood and Protein Drying Droplets"

_biomolecules, 2021, doi:10.3390/biom11020231_

Round 1

Reviewer 1 Report

      Dear Authors!        

    This manuscript “Temperature and Concentration Dependence of Human Whole Blood and Protein Drying Droplets” (Anusuya Pal , Amalesh Gope and Germano Iannacchione) provides experimental results of different pattern formation in the complex and in the simplest bio-colloidal droplet during the drying process. Whole human blood is taken as a complex system. As a simple -  aqueous solution of the globular protein, BSA. The biophysical problems of drying in such systems are discussed.

       This manuscript presents the results of continued research by the authors in this direction. The studies presented in the work are of scientific interest.

     I have the following remarks on the manuscript.

1) In the submitted manuscript, some of the figures / their fragments overlap with the figures presented in the published article [Reference 22]: Pal, A., Gope, A., Obayemi, J.D. et al. Concentration-driven phase transition and self-assembly in drying droplets of diluting whole blood. Sci Rep 1018908 (2020). https://doi.org/10.1038/s41598-020-76082-6.  Similar are  also model representations in both works. Problems that are presented in the texts of the submitted manuscript and in the published article intersect.

       Therefore, I consider  that in the Discussion section, the authors of this manuscript submitted for review, must necessarily include points in which they must conduct a detailed comparison of the material presented in their article in the  Scientific Reports and in the represented manuscript.

2) The authors consider the hierarchy of the behavior of the system components during droplet drying and argue that the self-interacting mechanism of large number of components will be observed for other complex multi-component systems, not specifically to the blood. Examples of such systems and situations would be desirable. Please, indicate the significance of this approach for physics and biology. Since the authors study the blood, it was interesting to know how to apply such approach to physiology and medicine.

3) The authors dilute blood with distilled water and discuss the interaction of blood components. When blood diluted with distilled water, hemolysis of erythrocytes will occur, in the solution will appear: RBC ghosts, fragments of the spectrin matrix, which also consists of protein structures, as well as free hemoglobin molecules. I would like the authors  to discuss how these components  behave,  what contribution they make to the  hierarchy pattern during drying process.

          It is possible to change the ratio of blood components  and plasma proteins not by diluting the blood with water, but by decreasing the concentration of erythrocytes using the centrifugation method. This would make it possible to study the hierarchy of the components of the bio-system upon drying, without violating their integrity.

4) Extra parenthesis in the legend Fig. 15. ….   20 wt%) in ….

This manuscript would be useful for publication after the Major Revision.

Reviewer 2 Report

The manuscript is well written and supported by data. The figures and graphs are made seriously with errors bars. The analysis is well done.

The reviewer found that this manuscript is in the direct continuation of similar work published recently in Scientific Reports - Ref. 22. Indeed, the authors try to answer to questions raised in ref. 22 in this manuscript. Here the influence of temperature is shown but does not show significative changes in the patterns, just a few changes due to the drying rate, as indicated in the conclusion. Can the authors emphasize on the utility of increasing substrate temperature for now and future studies?

The reviewer see a lot of observations and correlations in this manuscript without modeling and understanding of the involved mechanisms. For example in Fig. 13.1, there is no tentative to model w/R as a function of phi_pb. Is there modeling under preparation for future publications?

Round 2

Reviewer 1 Report

The manuscript can be published in this Journal Biomolecules